# Longitudinal Transcription Profiling of Bladder Cancers Dictate the Response to BCG Treatment and Disease Progression

**DOI:** 10.3390/ijms25010144

**Published:** 2023-12-21

**Authors:** Seo-Young Lee, Yun-Hee Lee, Tae-Min Kim, U-Syn Ha

**Affiliations:** 1Department of Medical Informatics, College of Medicine, The Catholic University of Korea, Seoul 03083, Republic of Korea; seoyoung64@catholic.ac.kr; 2Cancer Research Institute, College of Medicine, The Catholic University of Korea, Seoul 03083, Republic of Korea; 3Department of Urology, College of Medicine, The Catholic University of Korea, Seoul 03083, Republic of Korea; eyh900@catholic.ac.kr; 4Department of Biomedicine & Health Sciences, Graduate School, The Catholic University of Korea, Seoul 03083, Republic of Korea

**Keywords:** bladder cancers, BCG treatment, transcriptome, non-negative matrix factorization, gene set enrichment analysis

## Abstract

Although the intravesical instillation of Bacillus Calmette-Guerin (BCG) is widely used as adjuvant treatment for nonmuscle-invasive bladder cancers, the clinical benefit is variable across patients, and the molecular mechanisms underlying the sensitivity to BCG administration and disease progression are poorly understood. To establish the molecular signatures that predict the responsiveness and disease progression of bladder cancers treated with BCG, we performed transcriptome sequencing (RNA-seq) for 13 treatment-naïve and 22 post-treatment specimens obtained from 14 bladder cancer patients. To overcome disease heterogeneity, we used non-negative matrix factorization to identify the latent molecular features associated with drug responsiveness and disease progression. At least 12 molecular features were present, among which the immune-related feature was associated with drug responsiveness, indicating that pre-treatment anti-cancer immunity might dictate BCG responsiveness. We also identified disease progression-associated molecular features indicative of elevated cellular proliferation in post-treatment specimens. The progression-associated molecular features were validated in an extended cohort of BCG-treated bladder cancers. Our study advances understanding of the molecular mechanisms of BCG activity in bladder cancers and provides clinically relevant gene markers for evaluating and monitoring patients.

## 1. Introduction

Bladder cancers rank among the most prevalent human malignancies globally and are listed as the tenth most common cancers, contributing to a considerable number of cancer-related deaths annually [1,2,3]. While the prognosis of the disease depends on many clinicopathological factors, including tumor grade, size and stages with notable heterogeneity in clinical outcomes, muscle invasion is the primary indicator to classify bladder cancers into non-muscle-invasive (NMIBC) and muscle-invasive bladder cancers (MIBC) [4,5]. The aim of NMIBC treatment is to preserve bladders with endoscopic tumor resections as compared to radical cystectomy as the primary treatment option for MIBC. Despite better clinical outcomes of NIMBC compared to MIBC, the disease is frequently associated with disease recurrence and progression into muscle-invasion disease [6,7,8].

The use of intravesical Bacillus Calmette-Guerin (BCG) therapy, which involves a live attenuated strain of *Mycobacterium bovis*, originally developed for tuberculosis vaccination, marks a significant milestone in cancer treatment [9]. Introduced in 1976 as the first form of anti-cancer immunotherapy, it was specifically approved by the FDA in 1990 for treating nonmuscle-invasive bladder cancers (NMIBC) [10]. BCG immunotherapy has since been the standard adjuvant treatment for NMIBC, showing durable responses, particularly in high- and moderate-risk cases. However, 20–40% of patients do not respond to BCG treatment, presenting a significant challenge due to the high rates of disease recurrence and refractoriness [11]. The relatively high rates of disease refractoriness and the high rates of disease recurrence with BCG remain major hurdles in NMIBC treatment; non-responsive patients are generally recommended for radical cystectomy and other more aggressive treatment options [12]. To improve BCG treatment outcomes, it is crucial to understand the molecular mechanisms of BCG’s impact on anti-tumor immunity. While the immunological aspects of BCG therapy have been extensively reviewed, the exact mechanisms remain partially understood [13,14]. Both cell line-based experiments and human studies have demonstrated that BCG instillation upregulates various cytokines and chemokines, crucial for recruiting immune cells that bolster anti-tumor immunity. This involves both innate and adaptive immune responses [14]. However, a complete understanding of how BCG interacts with immune and stromal cells in the bladder cancer context is still lacking, including the absence of biomarkers to predict treatment responsiveness [14].

The heterogeneity of bladder cancer poses additional challenges in clinical evaluation and treatment decision-making. One attempt to address the tumor heterogeneity is the establishment of consensus subtypes based on transcriptomes, e.g., MIBC and NMIBC subtypes with distinct clinical behaviors and patient outcomes of bladder cancers [15,16]. For example, Hedegaard et al. utilized consensus clustering to predict the disease’s progression during BCG treatment, and they defined three distinct classes of patients [17]. Moreover, advanced methods like deconvolution have been valuable in dissecting the complex, heterogeneous data in cancer research. Meng et al., for instance, employed non-negative matrix factorization (NMF) to classify patients based on their response to BCG, focusing on immune-related signals [18]. Such approaches highlight the potential of transcriptome data in predicting molecular and clinical behaviors, including drug sensitivity and disease progression in bladder cancers treated with BCG. This emerging field of research holds promise for better understanding and eventually improving the outcomes of BCG treatment in bladder cancer. Thus, with proper evaluation, transcriptome data, which harbor tumor-intrinsic and -extrinsic signals, could potentially predict molecular and clinical behavior, including drug sensitivity and disease progression in BCG-treated bladder cancers.

In our research, we analyzed comprehensive transcriptome sequencing data from bladder cancer patients who experienced varying outcomes following BCG treatment. To address the inherent heterogeneity in this bulk-level transcriptomic data, we utilized NMF as an unsupervised deconvolution method to uncover hidden features. Our analysis revealed an immune-related molecular feature in treatment-naïve cases that appears to influence patient responsiveness to BCG therapy. This suggests that the immune context is a critical determinant of sensitivity to BCG treatment. Additionally, we found that the upregulation of genes related to the cell cycle and proliferation correlates with disease progression post-BCG treatment. To confirm the relevance of these BCG-progression features, we validated them in a separate group of patients who had undergone BCG therapy for bladder cancer, thereby establishing their clinical significance. Our study contributes to a deeper understanding of the molecular factors that impact both the response to BCG therapy and the progression of the disease following treatment.

## 2. Results

### 2.1. Patient Demographics

Fourteen patients who underwent endoscopic tumor resection and also received intravesical BCG instillation as adjuvant therapy were enrolled in this study. The average age of onset was 73.8 years, and 79.6% (*n* = 11) of the patients were male. Clinicopathological data for the patients are available in Figure 1A. To illustrate the clinical course of individual patients, we used a swimmer plot showing the type and duration of treatments, including BCG instillations (Figure 1A). To summarize the variable clinical outcomes and courses of individual patients, we used the following clinical annotations. The ‘stable/progression’ class indicates whether patients experienced disease progression, and the ‘success/fail’ class indicates whether patients initially responded to BCG instillation. Overall, four patients (28.6%) experienced disease progression, and three patients (21.4%) showed responsiveness to BCG treatment. During follow-up, one patient (P6) died with lymph node and bone metastases. The remaining 14 patients were alive at the time of writing. Transcriptome sequencing was performed on the specimens obtained before and after treatment (13 pre-BCG/pre-treatment specimens and 22 post-BCG/post-treatment specimens) (Figure 1B). While the segregation of the data is predominantly driven by cases, suggesting a substantial level of inter-tumoral heterogeneity, the clinical features (drug response and disease progression) were not clearly segregated in the results. Even though gene expression was not significantly divided by the clinical features, we found that it was divided by the immune and stromal scores estimated from the transcriptome data. For example, patients who showed BCG responsiveness also showed high immune scores and stromal scores, whereas patients with disease progression showed relatively low immune scores and stromal scores. Thus, we assumed that the transcriptional features associated with the phenotypes of interest (BCG responsiveness and disease progression) might be present, albeit to a low extent.

### 2.2. Deconvolution of Latent Transcriptional Features in Bladder Cancers

To cope with tumor heterogeneity, we used NMF-based deconvolution. The objective of NMF is to decompose a bulk-level expression data matrix into a non-negative linear combination of molecular features corresponding to latent molecular behaviors (NMF features) and their respective coefficients (the sample-level abundance of NMF features) [19]. The plot of cophenetic correlation coefficients exhibited a gradual decrease within the range of K (2 to 20, Appendix A), so we empirically determined the number of clusters (the K value) to be 12 and used that value to capture the molecular behaviors of the cohort. First, the coefficient matrix, which represents the relative abundance of the 12 identified NMF features (referred to as NMF1–NMF12 hereafter), was subjected to hierarchical clustering. The clustering heatmap of the NMF features is shown in the clinical presentation (Figure 2A). Although patient-level segregation is still present, we note that several NMF features are associated with BCG responsiveness and disease progression, e.g., NMF1 is relatively upregulated in BCG responders (P2, P4, and P13), and patients with disease progression showed upregulation of NMF4 and NMF11. In addition, NMF1 exhibited high immune scores and stromal scores, indicative of elevated levels of TME infiltration. We next examined the associations between the NMF-derived features and the mSigDB Hallmark gene sets (Figure 2B). NMF1 was observed in proximity to the molecular functions associated with immunity (e.g., “IL6-JAK-STAT3 signaling” and “inflammatory response”), as was NMF2; however, NMF1 was also associated with stromal-related functionalities (e.g., “epithelial-to-mesenchymal transformation (EMT)” and “myogenesis”). The NMF features often showed co-segregation, e.g., the coefficients of NMF features 3, 5, 9 and 10 were commonly enriched for the molecular functions involved in the metabolism of oxidative phosphorylation and glycolysis. Similar observations were made for the gene sets of the curated molecular functions (c2cp, c5 and the hallmark gene ontology sets for molecular functions and biological processes, Appendix A). The top ten differentially expressed marker genes are shown in Figure 2C. The NMF1-enriched genes include *HLA-DRA*, *HLA-DRB1* and *HLA-DRB5*, which are major histocompatibility genes, and *CD74*. *CTSB*, which encodes cathepsin B, is also observed in NMF1 and is known to be associated with the remodeling of the extracellular matrix (ECM), indicating the potential stromal features of NMF1 [20]. Various chemokine molecules, *CCL18, CCL19, CCL21, CCL22* and *CXCL13*, were observed in NMF2, indicating its association with immune activity. The gene markers for NMF4 were often associated with ECM-related functions. For example, a high expression of *CST1* is known as an unfavorable prognostic marker across tumors and experimentally induced EMT in lung cancers [21]. *COMP* encodes noncollagenous ECM profiles and might promote EMT in cancer cells [22]. In addition, anti-cancer immune-related genes were observed, e.g., *SERPINB4* is highly expressed in cancer cells and inactivates anti-tumor enzymes such as granzyme M [23].

### 2.3. Clinical Concordance of NMF Features

To select the NMF features that are clinically relevant in the BCG context, we examined the level of association between each of the 12 NMF features and BCG responsiveness and disease progression. The feature-level associations with BCG responsiveness and disease progression are shown for the total dataset and the pre- and post-treatment datasets (Figure 3A). In the total dataset (pre- and post-treatment specimens), NMF1, NMF2 and NMF4 showed significant associations with BCG responsiveness and disease progression (*p* = 0.0052, 0.0026, and 0.0001, K–S test), respectively. Those associations were found to be largely attributable to the pre-treatment samples (NMF1 with BCG responsiveness, *p* = 0.0280) and post-treatment specimens (NMF2 and NMF4 with disease progression, *p* = 0.0003 and 0.0001, respectively). Sample-level associations were further demonstrated (Figure 3B), indicating that BCG responders are significantly enriched in NMF1 pre-treatment (*p* = 0.0052, K–S test) and tumors with disease progression are significantly high in NMF4 and NMF2 post-treatment (*p* = 0.0026 and *p* = 0.0001, respectively, K–S test). However, the NMF2-high cases were relatively depleted in the samples with disease progression, which suggests that NMF2 might represent favorable risk factors (Appendix A). The MCP results from the pre-treatment samples indicate that the responsive cases are associated with a higher immune cell score than the non-responsive cases, which suggests the existence of a distinct BCG-responsive immune context that can be identified before treatment. The immune scores were largely comparable between post-treatment samples with and without disease progression. The cellular progression and cell cycle levels did not differ substantially with respect to disease progression (Figure 3C).

### 2.4. Validation of Features Predicting Disease Progression after BGC

We downloaded the transcriptome data from bladder cancers with a BCG treatment history from an independent resource (GSE199471 cohort). Among the data, we selected 14 specimens treated with BCG and used them as validation data for the NMF4-based prediction of disease progression. We selected 71 signature genes that represent the activity of NMF4 and used the median of the signature gene expression to sort the cases. The extent of enrichment in cases with disease progression in the ranked list was statistically significant (NMF4 with BCG progression, *p* = 0.022, K–S test) (Figure 4A). We further examined the expression levels of individual genes belonging to the progression features according to the literature (Figure 4B). Overall, upregulation of the selected genes was observed in cases with disease progression.

## 3. Discussion

In this study, we used longitudinal biopsies from BCG-treated bladder cancers and found (1) a strong correlation between the expression of immune-related pathways and BCG responsiveness in treatment-naïve bladder cancers and (2) a relationship between the activity of cell cycle-related genes and disease progression following BCG treatment. It has been widely recognized that BCG treatment triggers a localized immune response in both cancer cells and surrounding cells; however, the response to BCG treatment can vary among individuals. Factors such as the immunogenicity of the tumor, the overall immune competence of the patient and specific characteristics of the tumor microenvironment can influence the efficacy of BCG therapy. Among the genetic factors, specific *HLA* (human leukocyte antigen) types have been reported to be associated with the variable response to BCG treatment [21]. A high level of *HLA-DRA* and *HLA-DRB*, which are MHC class II molecules, emerged as potential markers for NMF1, indicating BCG responsiveness. According to a study [24], BCG-infected antigen-presenting cells, such as macrophages and dendritic cells, demonstrate a decrease in MHC class II expression. Additionally, the level of MHC class II expression has been found to be linked to the effectiveness of BCG vaccination against tuberculosis [24]. Therefore, we assume that the level of MHC class II expression could indicate the efficacy of BCG-induced anti-tumor immunity and be used to predict BCG responsiveness in bladder cancers. It is noteworthy that the immune-related NMF1 predicts BCG responsiveness before BCG treatment but not after it, perhaps because BCG induction leads to a substantial immune reaction regardless of BCG responsiveness. Nevertheless, the pre-treatment immunologic context, which can be monitored by a latent factor representing TME infiltration, including MHC class II expression, could potentially provoke distinct clinical outcomes. Furthermore, NMF2, which is largely enriched with various types of cytokines and chemokines, is distinct from NMF1 and does not predict BCG responsiveness. Thus, only some aspects of pre-existing anti-cancer immunity in bladder cancers before BCG treatment can predict the overall response to BCG. The small number of cases in our cohort should be taken into account when analyzing our results, and a large cohort might be necessary to further delineate the specific immune response that determines BCG responsiveness before BCG treatment.

Traditionally, the analysis of cell cycle models and behaviors has been predominantly conducted through flow cytometry. However, recent advancements have shown that in silico methods are also highly effective, particularly in the fields of drug discovery and understanding drug mechanisms of action. In our study, we employed a simplified approach focusing on the signature-based assessment of cell doubling times and levels of cellular proliferation [11,25]. This method represents a significant shift from conventional laboratory techniques, leveraging computational analysis to gain insights into cellular dynamics. Our hypothesis posits that effective anti-cancer therapies, including BCG instillation for bladder cancer, should result in the downregulation of cell cycles and cellular proliferation. In line with this, we discovered that a specific feature identified by our analysis, NMF4, which represents aspects of the cell cycle and proliferation, correlates with disease progression in bladder cancer patients post-BCG treatment. This finding suggests that NMF4 could potentially serve as a biomarker for the individual effectiveness of BCG therapy, highlighting its role in disease progression. The prospect of using NMF4 to predict disease progression or relapse is an exciting avenue for future research, especially if validated in larger, independent datasets. Such validation could solidify its role as a reliable marker in clinical settings. However, it is crucial to note that our study’s conclusions are currently limited by the small number of cases examined and the limited size of the validation dataset. These constraints underline the need for further comprehensive studies to fully establish the broad applicability and reliability of this marker in the context of bladder cancer treatment. This expansion of research is essential for confirming the potential of NMF4 as a predictive tool in clinical practice, offering a more nuanced understanding of individual patient responses to BCG therapy.

The unique aspect of our study lies in its focus on delivering clinically pertinent insights drawn from the longitudinal RNA expression data obtained during the course of BCG treatment for bladder cancer. This approach is distinguished by previous studies that often offered ‘snapshots’ of molecular-genetic information. Using longitudinal approaches, we have been able to identify specific feature-level markers that are both predictive and prognostic in nature. Firstly, they enable us to pinpoint patients who are likely to benefit from BCG therapy. This is particularly important given the variability in patient responses to BCG, allowing for a more personalized approach to treatment. Secondly, these markers can identify individuals at a high risk of disease recurrence post-BCG treatment. Identifying such patients is essential, as it suggests a need for alternative therapeutic strategies such as cystectomy, given their higher risk profile. The insights gained from this study have practical implications for the precision medicine of bladder cancer treatment. Knowledge about molecular vulnerabilities in NMIBC, when integrating these feature-level markers into treatment planning, may inform clinicians to make more informed decisions to tailor therapies to individual patient profiles, potentially improving outcomes and reducing the likelihood of disease recurrence.

## 4. Materials and Methods

### 4.1. Patient Cohort

Fourteen patients admitted to Seoul St. Mary’s Hospital (College of Medicine, Catholic University of Korea, Seoul, Republic of Korea) between May 2016 and July 2021 were enrolled in this study. The study was performed with the approval of the Institutional Review Board (IRB no KC19SESI0223). Informed consent was obtained from each of the enrolled patients. Surgical specimens were collected during endoscopic resections and subsequently paraffin-embedded. Patients’ clinical responses (drug response and disease progression) were evaluated by an expert review of the medical charts.

### 4.2. Transcriptome Sequencing and Preprocessing

For transcriptome sequencing, RNA from formalin-fixed paraffin-embedded tumor tissue was extracted according to the manufacturer’s instructions. We used the SureSelectXT RNA Direct Reagent Kit (Agilent, Santa Clara, CA, USA) to construct sequencing libraries. The preprocessing of RNA sequencing data was conducted according to the NCI Genomic Data Commons workflow [26]. The fastq files were subject to QC by FastQC (Version 0.11.8) [27], and sequencing quality reports were generated using MultiQC (version 1.12) [28]. All the fastq files (quality score over Q35) were further processed by removing adapter sequences and low-quality bases using fastp (version 0.19.5) [29]. RNA sequencing reads in 100 bp paired-end mode were mapped to the GRCh38 human reference genome using STAR aligner in the 2PASS pipeline (v2.7.10a) [30]. The gene expression levels were estimated as read counts using RSEM (RNA-Seq by Expectation-Maximization) (v1.3.0) [31]. For expression analysis, the read counts were converted into transcript-per-million (TPM) values and were subject to subsequent analyses.

### 4.3. NMF Clustering and Extraction of Signature Genes

Hierarchical clustering for highly variable genes was performed using the median absolute deviation. To discover latent genetic profiles, we conducted NMF (R package version 0.25) analyses [31] for the entire dataset, i.e., the expression profiles of 13 treatment-naïve and 22 post-treatment transcriptomes. Cophenetic scores were used to determine the optimum number of latent features (K numbers), tested in the range of 2 to 20. As a result, the 35 treatment-naïve and post-treatment specimens were subjected to NMF-based deconvolution with K = 12. The NMF results, two matrices representing the latent genetic behaviors (basis) and their sample-level abundances (coef), were further analyzed. For the binary phenotypes of interest (e.g., drug responsiveness and disease progression), we used the Kolmogorov–Smirnov (K–S) test to estimate the statistical significance between two distributions, i.e., the coefficient difference between BCG-stable vs. -progression [31]. For binary phenotypes of interest (drug responsiveness and disease progression), we used the Kolmogorov–Smirnov (K–S) test to estimate the statistical significance between distributions [31]. Signature genes representing each NMF basis were extracted using fold change. All statistical analyses were performed using R (version 4.2.2).

### 4.4. Functional Analysis

To infer the molecular function of the latent NMF features, we used a gene set enrichment analysis (GSVA package version 3.16) to calculate each specimen’s enrichment scores for functional terms (Hallmark gene sets in the MSigDB database) [32]. To visualize the relationship between molecular functions and NMF-driven molecular features, a correlative map was generated, integrating the sample-level pathway activity and feature abundance. For microenvironment profiling, we used microenvironment (TME) cell population (MCP) counters that estimate the cellular abundance of immune and stromal cell groups from bulk-tumor transcriptomes [33]. The correlation map and other data matrices were visualized using a heatmap (version 1.10.12) [34].

### 4.5. Validation Data

For validation, we obtained transcriptome data for 14 high-risk NMIBC bladder cancers treated with BCG immunotherapy (11 patients with disease progression) from an independent resource (GEO accession: GSE199471 cohort) [35]. We used those 14 specimens to test whether our progression-related feature (NMF4) could distinguish patients who experienced disease progression from those who did not. The signature genes of the individual features were used to estimate the feature-wise scores as the mean of gene expression. The patients were sorted in order of their signature scores for each NMF feature, and the level of significance of enrichment was estimated by the K–S test.

### 4.6. Data Availability

The BCG Bladder Cancer RNA-seq dataset is available to download from the GEO repository through the accession name GSE244895. The BCG bladder validation dataset is available in the GEO database (GSE199471).

## Figures and Tables

**Figure 1 ijms-25-00144-f001:**
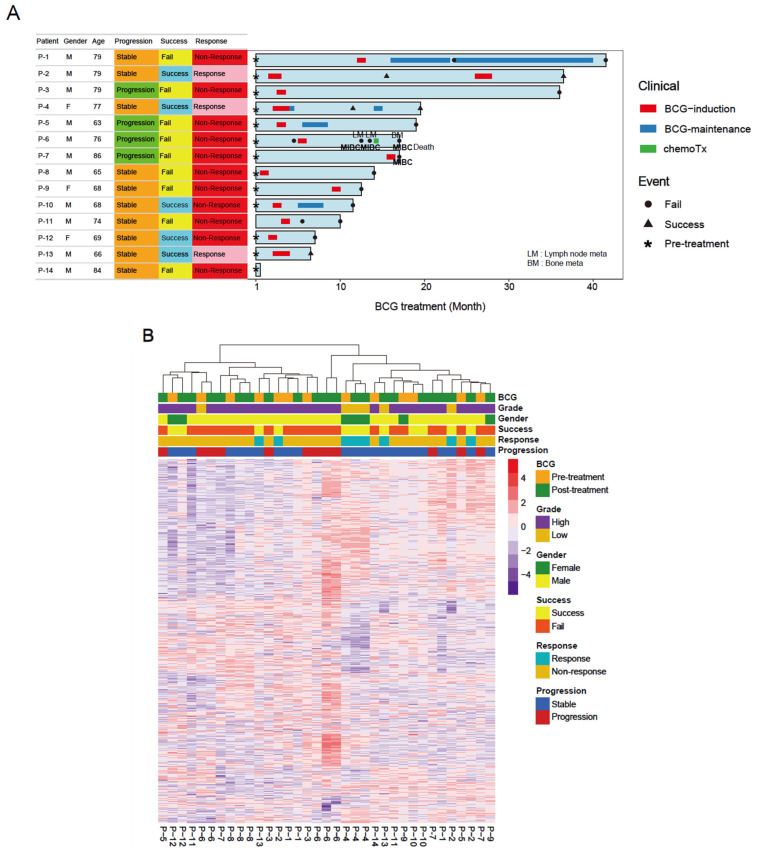
The clinical course of 14 patients and transcriptional heterogeneity. (**A**) A swimmer plot representing the clinical course of 14 patients with bladder cancer who were treated with BCG. Three types of clinical annotation are shown, along with the type and duration of treatment (BCG induction/maintenance and chemotherapy). (**B**) Hierarchical clustering of the bulk-level transcriptome dataset, largely segregated by sample.

**Figure 2 ijms-25-00144-f002:**
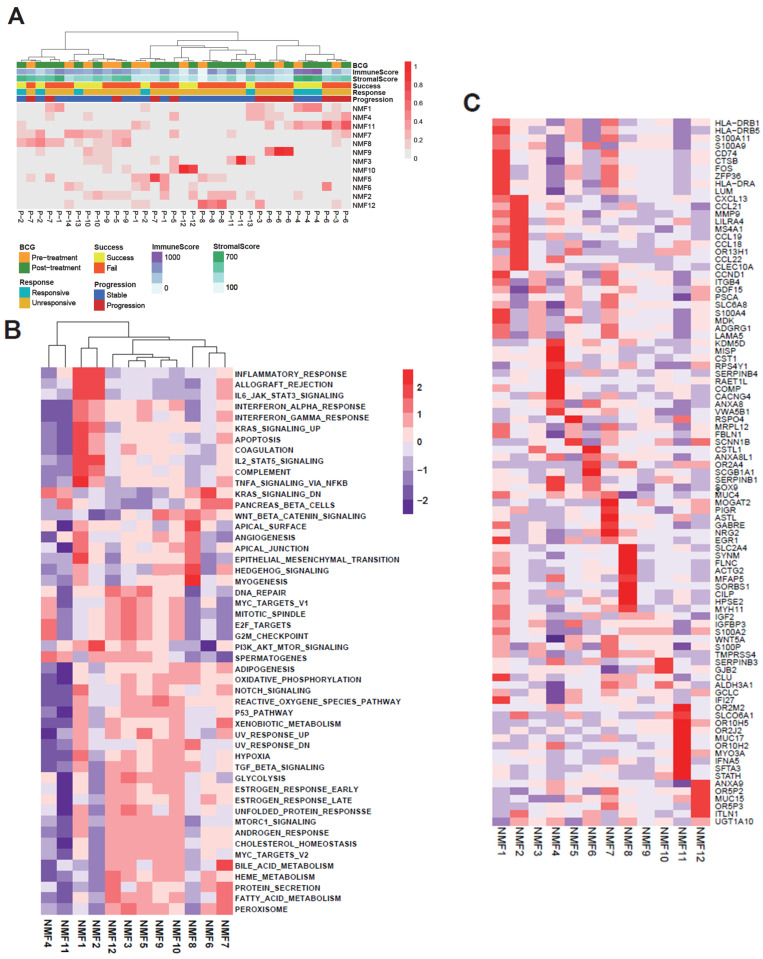
Factorized matrices W (basis) and H (coefficient) from the top 5000 genes used to characterize the NMF features. (**A**) Heatmap of the coefficient value between each NMF feature and the clinical course. (**B**) Heatmap of Pearson coefficient correlation values between NMF features and the Hallmark pathway values of basis. (**C**) Top 10 genes from the top 10 pathways in the classical pathway and the GO terms BP and MF for each NMF feature.

**Figure 3 ijms-25-00144-f003:**
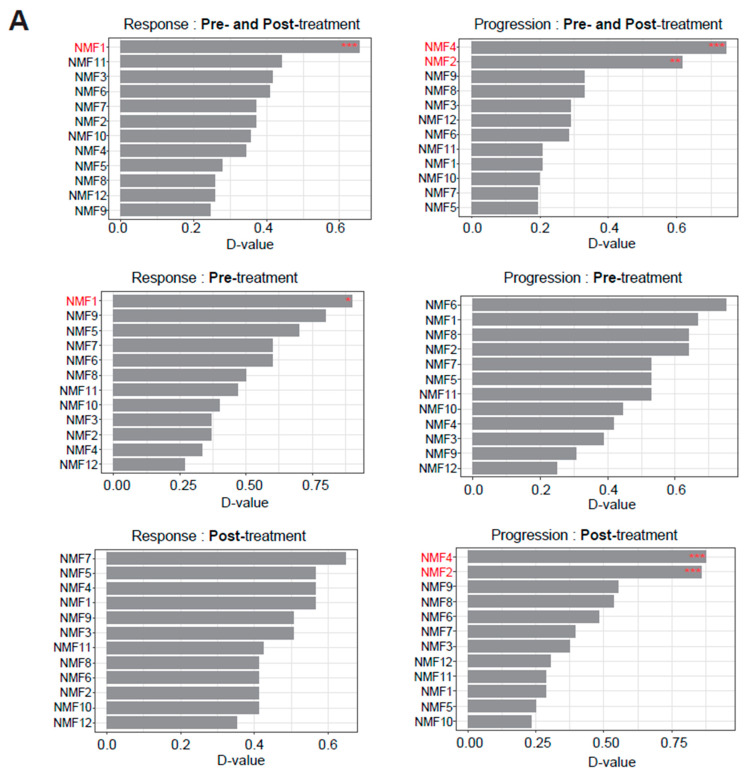
Identified NMF features related to the clinical course, with statistical significance and their biological meanings. (**A**) Kolmogorov–Smirnov (K–S test) D-values and *p*-values for the defined features associated with the clinical course. (**B**) Ranking of NMF1 and NMF4 coefficients in pre- and post-treatment samples. (**C**) MCP-counter result for NMF1 and NMF4 in pre- and post-treatment samples. K–S test, * *p* < 0.05, ** *p* < 0.01, *** *p* < 0.001.

**Figure 4 ijms-25-00144-f004:**
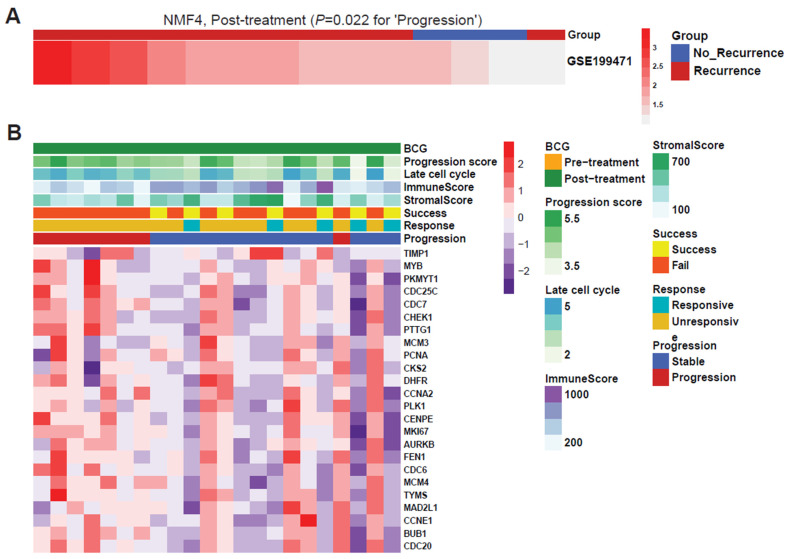
Confirmation that NMF4 can predict progression samples in the GSE199471 validation cohort and 24 progression signature genes. (**A**) Validation of NMF4 using the GSE199471 cohort. (**B**) Annotated gene expression matrix for 24 progression-signature genes ranked by their coefficient values.

## Data Availability

The BCG Bladder Cancer RNA-seq dataset is available to download from the GEO repository through the accession name GSE244895. The BCG bladder validation dataset is available in the GEO database (GSE199471).

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
