# Peer review of "Longitudinal Transcription Profiling of Bladder Cancers Dictate the Response to BCG Treatment and Disease Progression"

_ijms, 2023, doi:10.3390/ijms25010144_

Round 1
Reviewer 1 Report
Comments and Suggestions for Authors
Title: Longitudinal Transcription Profiling of Bladder Cancers dictate the Response to BCG Treatment and Disease Progression
Authors: Seo-Young Lee, Yun-Hee Lee, Tae-Min Kim and U-Syn Ha
Affiliations: Department of Urology, College of Medicine, The Catholic University of Korea, Seoul, Republic of Korea
Purpose: To establish the molecular signatures that predict responsiveness and disease progression of bladder cancers treated with intravesical BCG.
Type of Study: Prospective clinical study conducted at Seoul St. Mary’s Hospital College between May 2016-July 2012. TURBT specimens were processed as usual, and paraffin embedded tissue was used for RNA sequencing.
Findings: The authors found that gene expression was divided by the immune and stromal scores estimated from the transcriptome data. Patients that showed BCG responsiveness also showed high immune scores and stromal scores, whereas patients with disease progression showed relatively low immune scores and stromal scores. The tumor microenvironment cell population results from the pre-treatment samples indicate that the responsive cases are associated with a higher immune cell score than the non-responsive cases, suggesting the existence of a distinct BCG-responsive immune context that can be identified before treatment. This separation was lost post BCG. Surprisingly, the NMF2 population, which is largely enriched with various types of cytokines and chemokines, does not predict BCG responsiveness. The authors make the assumption that cellular proliferation should be down with BCG therapy, however if they are analyzing the whole cellular environment with infiltrating immune cells this may not be the case. Co-stimulatory markers are not examined/reported.
Minor Concerns: English as second language. Typo and syntax errors.
Major Concerns: Small sample size is analyzed and presented. Figures in preprint manuscript are not sharp (fuzzy)-poor quality and resolution.
Line by Line Review:
Line 15; “To establish molecular signatures that predict the responsiveness and disease progression of bladder cancers treated with BCG. We performed transcriptome sequencing (RNA-seq) for 13 treatment-naïve and 22 post-treatment specimens obtained from 14 bladder cancer patients.” Incomplete sentence – suggest ‘To establish molecular signatures that predict the responsiveness and disease progression of bladder cancers treated with BCG, we performed transcriptome sequencing (RNA-seq) for 13 treatment-naïve and 22 post-treatment specimens obtained from 14 bladder cancer patients.’
Line 52; “Thus with proper evaluation, transcriptome data, which harbor tumor-intrinsic and -extrinsic signals, can predict molecular and clinical behavior, including drug sensitivity and disease progression in BCG-treated bladder cancers. Syntax error – suggest ‘Thus with proper evaluation, transcriptome data, which harbor tumor-intrinsic and -extrinsic signals, could potentially predict molecular and clinical behavior, including drug sensitivity and disease progression in BCG-treated bladder cancers.’
Line 195; “In the total dataset (pre- and post-treatment specimens), NMF1 and NMF2 and 4 showed significant associations with BCG responsiveness and disease progression (P = 0.0052, 0.0026, and 0.0001, K-S test), respectively.” Typo- suggest ‘In the total dataset (pre- and post-treatment specimens), NMF1 and NMF2 and NMF4 showed significant associations with BCG responsiveness and disease progression (P = 0.0052, 0.0026, and 0.0001, K-S test), respectively.
Line 243; “Additionally, the level of MHC class II expression has been found to be linked to the effectiveness of BCG vaccination against tuberculosis.” Suggest that you provide a reference to this statement as you have not provided data.
Comments on the Quality of English LanguageSee comments above.
Reviewer 2 Report
Comments and Suggestions for Authors
Bladder cancer (BCa) is the most frequent malignant carcinoma of the genitourinary tract, being the 10th tumor for incidence when considering both sexes, and the 7th considering only the male population.
Intravesical administration of Bacillus Calmette-Guerin (BCG) is the gold standard treatment in the adjuvant setting for high-grade/G3 non-muscle invasive bladder cancer (NMIBC) inducing a durable immune response in the tumoral and peri-tumoral tissues.
NMIBC patients treated with 1–3 years of maintenance BCG are among the highest risk for disease progression, with 1- and 5-year rates of 11.4% and 19.8%, respectively.
To address these issues and improve clinical outcomes in this cohort of patients, the authors aimed to investigate the molecular mechanisms that underlie the antitumor immunity effects of BCG and affect responsiveness to and disease progression after BCG treatment.
The authors should be congratulated for the interesting topic discussed.
I believe that the study has sufficient merit to be considered for publication, although major revisions are required.
1. Methods and methodology are robust.
2. Results and conclusions are well presented.
3. Tables and graphics are clearly described.
4. It is worth mentioning the current progress done in scientific and clinical fields when it comes to predicting tumoral recurrence and BCG unresponsiveness. I suggest providing more detailed information about it. These two papers offer a valid solution for the purpose (https://doi.org/10.3390%2Fdiagnostics12030586, https://doi.org/10.1016/j.urolonc.2022.05.016 ). A lecture is suggested.
Comments on the Quality of English LanguageMinor editing.
Round 2
Reviewer 2 Report
Comments and Suggestions for Authors
Authors answered all comments and suggestions.
Comments on the Quality of English LanguageMinor editing.